# Collagen breaks at weak sacrificial bonds taming its mechanoradicals

Benedikt Rennekamp[1,2,3], Christoph Karfusehr ®[1,3,4], Markus Kurth[1,2], Aysecan Ünal[1,2,3], Debora Monego ®[1], Kai Riedmiller ®[1], Ganna Gryn'ova ®[1,2], David M. Hudson[5] & Frauke Gräter ®[1,2,3] ✉

Collagen is a force-bearing, hierarchical structural protein important to all connective tissue. In tendon collagen, high load even below macroscopic failure level creates mechanoradicals by homolytic bond scission, similar to polymers. The location and type of initial rupture sites critically decide on both the mechanical and chemical impact of these micro-ruptures on the tissue, but are yet to be explored. We here use scale-bridging simulations supported by gel electrophoresis and mass spectrometry to determine breakage points in collagen. We find collagen crosslinks, as opposed to the backbone, to harbor the weakest bonds, with one particular bond in trivalent crosslinks as the most dominant rupture site. We identify this bond as sacrificial, rupturing prior to other bonds while maintaining the material's integrity. Also, collagen's weak bonds funnel ruptures such that the potentially harmful mechanoradicals are readily stabilized. Our results suggest this unique failure mode of collagen to be tailored towards combatting an early onset of macroscopic failure and material ageing.

Collagen is the most abundant protein of our body and the major material of all connective tissue, from tendon to bone to skin. It is perpetually subjected to high mechanical loads, such as up to 90 MPa in stretched achilles tendon[1]. As recently discovered, under elastic deformations, tendon collagen I forms mechanoradicals originating from covalent bond rupture[2], just as in any other synthetic polymer[3,4]. However, in contrast to synthetic materials, collagen readily stabilizes the primary radicals on dihydroxy-phenylalanines (DOPAs), with these highly efficient radical-scavenging residues built into the collagen protein[2]. Where collagen I, and potentially other protein materials, rupture under mechanical load remains to be clarified. The microscopic failure mechanisms, however, define macroscopic mechanical properties such as ultimate strain and toughness. Further, the positions of the weakest links within the structure determine where and what types of mechanoradicals are formed and how they will propagate through and potentially damage the system until they get scavenged.

A tempting hypothesis for collagen is, therefore, that rupture occurs in the vicinity of DOPAs to rapidly stabilize the primary radical and thereby prevent uncontrolled radical reactions such as migration and recombination. Nailing down the rupture process of collagen, being the major substituent of tissues, is a key necessity for addressing tissue degradation and ageing, and for guiding tissue engineering.

The first determinant of rupture is the relative strength of the chemical bonds. Any protein backbone features three types of bonds: the C-N peptide bond, the $C_\alpha$-N bond, and the $C_\alpha$-C bond. Due to the $\pi$-conjugation across the peptide bond, this bond is an unlikely candidate for homolytic rupture. Which of the two single bonds, $C_\alpha$-N or $C_\alpha$-C, is the weaker link is less obvious. Previous studies suggest bond dissociation energies for these two bond types to be in a similar range[2,5], so the chemical environment matters. Collagen also features enzymatically derived crosslinks connecting the individual triple helices. A variety of crosslink chemistries has evolved, which can be

[1]Heidelberg Institute for Theoretical Studies, Schloss-Wolfsbrunnenweg 35, 69118 Heidelberg, Germany. [2]Interdisciplinary Center for Scientific Computing, Heidelberg University, INF 205, 69120 Heidelberg, Germany. [3]Max Planck School Matter to Life, Jahnstrasse 29, 69120 Heidelberg, Germany. [4]Physics Department and ZNN, Technical University Munich, Coulombwall 4a, 85748 Garching, Germany. [5]Department of Orthopaedics and Sports Medicine, University of Washington, Seattle, WA 98195, USA. ✉e-mail: frauke.graeter@h-its.org

categorized into divalent (or pre-mature) and trivalent (or mature) crosslinks. All of them are derived from lysines, and as a consequence, feature again single bonds of the C-C and C-N type (compare Fig. 1c). The thermodynamic differences in the homolytic scission of collagen's backbone and crosslink bonds are decisive for collagen mechanics but currently unknown.

The second determinant of rupture is how force distributes through the complex hierarchical structure of collagen (compare Fig. 1) and loads its chemical bonds. In collagen, $\alpha$-chains wind up to triple helices that are packed in a quasi-hexagonal shape in cross-section. They are braided into each other in a non-trivial way along the fibril axis, forming fibrils on a larger scale[6]. This three-dimensional structure, combined with the crosslinking in between the triple helices, results in a molecular force distribution network that critically decides on the rupture propensity of an individual bond. In the simplest model, the Bell-Evans model, a force acting on a bond exponentially increases its rate of rupture[7–9]. Several previous studies have suggested that crosslinks are likely to be rupture sites in collagen: Crosslinks have been reported as responding to stress and suggested as rupture candidates in smaller-scale molecular simulations[10] as well as in coarse-grained models[11]. We previously identified crosslinks as well as adjacent backbone bonds as the most strongly loaded links[2,12]. Finally, recent semi-empirical calculations identified $C_\alpha$-C bonds at X-positions (in particular of prolines) as rupture candidates[13], but also suggested one exemplary crosslink as another preferred rupture point. None of the previous studies allowed quantitative conclusions on specific bond rupture sites within the collagen fibril.

In the field of polymer mechanochemistry, selective bond rupture has been achieved by synthetically incorporating weak bonds into the polymer, as sacrificial bonds increase the toughness of the material[14], or as mechanophores to report on mechanical force[15,16]. Sacrificial bonds that so far have been observed in biological systems are non-covalent in nature, such as hydrogen bonds or ionic bonds in bone collagen[17,18]. If protein materials harness the virtue of mechanically weak covalent bonds as sacrificial bonds or mechanophore, is, to our knowledge, currently unknown. In collagen, selective bond scission could critically determine mechanical stability, further radical-mediated damage, and modes of mechano-sensing.

Here, we employ comprehensive and high-level quantum mechanical calculations, scale-bridging hybrid molecular dynamics simulations for reactive dynamics (KIMMDY[12]), and gel electrophoresis-coupled mass spectrometry (MS) of stressed collagen to assess collagen's mechanochemical rupture. Our computations identify crosslink bonds as weakest links and major rupture sites, in particular, $C_\alpha$-$C_\beta$ bonds, due to the stabilization of $C_\alpha$-centered radicals by the captodative effect. We observe a large concentration of ruptures of ~70% in these bonds, the two key factors of which are ~120 kJ mol$^{-1}$ lower bond dissociation energies of crosslink compared to backbone bonds, and the higher stress concentration in this region. As a result, most rupture sites are in direct vicinity to DOPA residues which can prevent uncontrolled radical reactions by rapid scavenging. However, there is strong competition with unspecific breakages in the vast amount of backbone bonds throughout the fibril. To still direct rupture into the comparably few crosslink bonds, these bonds have to be substantially weaker. SDS-PAGE coupled to MS supports altered collagen crosslinking in stressed tissue compared to control samples, as well as a widespread rupture within both $\alpha$1 and $\alpha$2-chains in collagen I, as evident from the redistribution of molecular masses in collagen upon mechanical stress. Our simulations find a particular bond in one of the two 'arms' of the trivalent crosslink Pyridinoline (PYD) and Deoxypyridinoline (DPD) to be particularly primed for rupture due to a stabilizing conjugation of the resulting radicals. Its rupture does not compromise collagen integrity and leads to local stress relaxation, a hallmark of a sacrificial bond.

Overall, we show both experimentally and computationally that the conjunction of topology and chemical structure of collagen by design funnels bond ruptures towards crosslinks and specific sacrificial bonds therein, very much akin to weak sacrificial bonds in synthetic polymers. Together with previous findings that potential sites of the radical scavenger DOPA are highly enriched around these regions[2], also across different tissue types[19], our results put forward collagen as a biomaterial with highly specific mechanochemical routes, a feature which likely prevents further unspecific damage by radicals.

## Results
### Crosslinks harbor the weakest bonds
To determine the stability of covalent bonds in collagen I and identify the thermodynamically weakest links, we performed high-accuracy QM calculations on the G4(MP2)-6X level of theory, which reaches an accuracy similar to that of G4 for thermochemical properties[20].

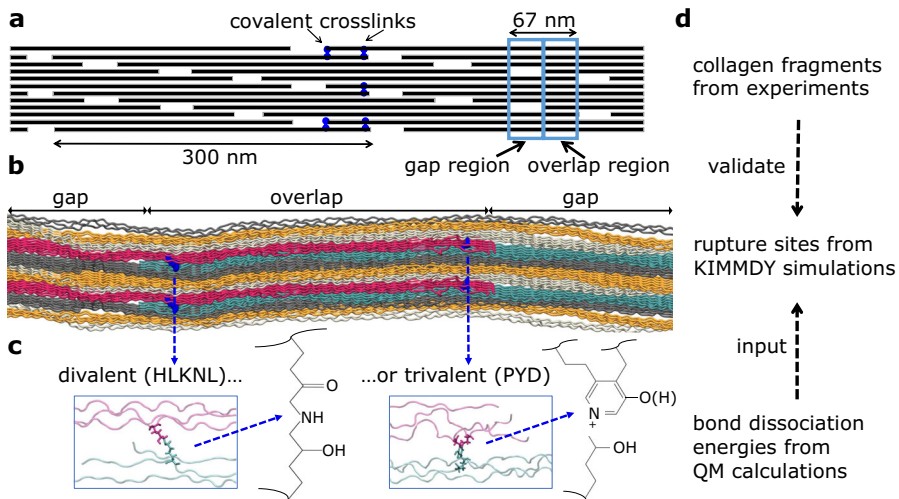

**Fig. 1 | Collagen structure, spanning multiple length scales, and our corresponding methods. a** 2D projection of the staggered arrangement of collagen triple helices, each 300 nm. This results in the typical overlap and gap regions of collagen, including the 3D braiding of triple helices. **b** Our atomistic model spanning one overlap (middle) and about one gap region (split into two parts) of collagen. **c** Zoom in on enzymatic crosslinks connecting the triple helices. Different chemistries (divalent or trivalent) are possible at these positions, for example, Hydroxylysino-keto-norleucine (HLKNL) or Pyridinoline (PYD). **d** Our workflow combines different methods as collagen spans multiple length scales.

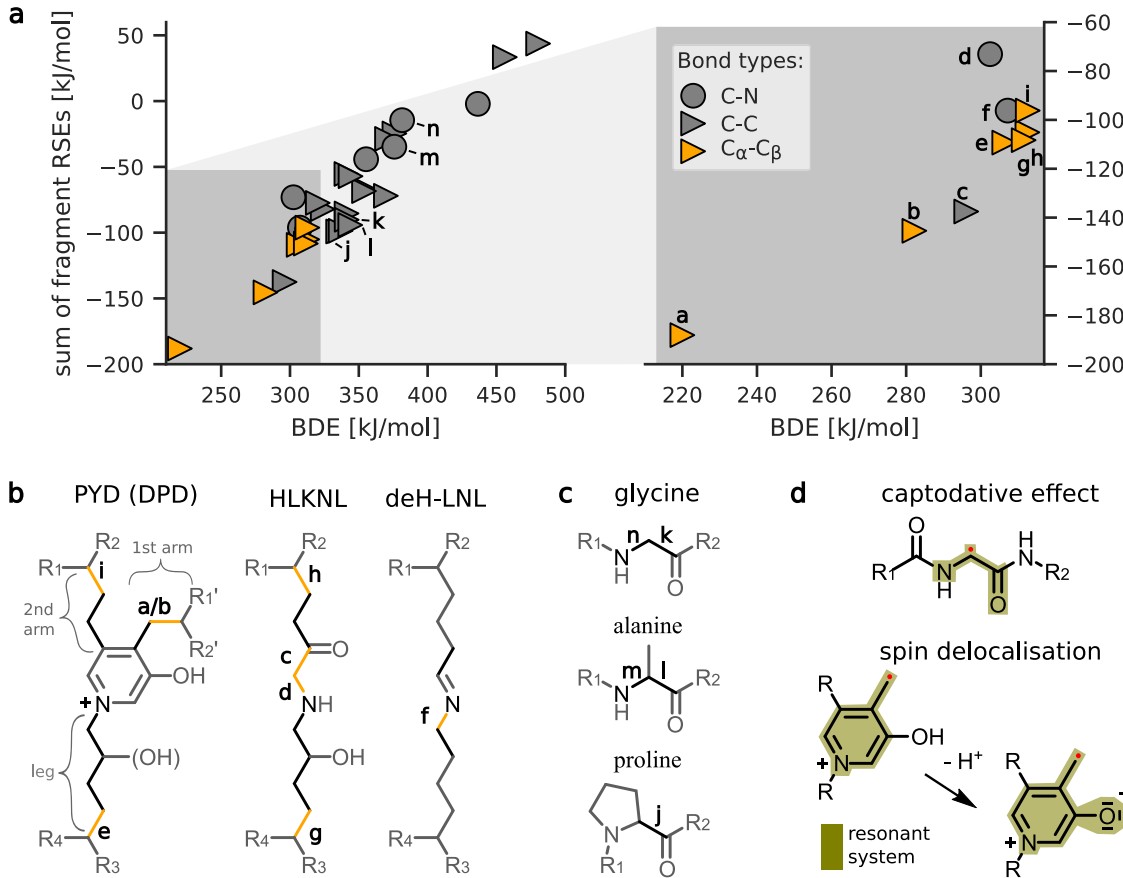

**Fig. 2 | Crosslinks harbor the weakest bonds in collagen type I. a** Correlation between the bond dissociation energies (BDEs) and summed radical stabilization energies (RSEs) of homolytically cleaved bonds and formed radicals. All molecular structures used for calculations can be found in the SI. Bonds with BDEs <315 kJ mol⁻¹ are shown in more detail on the right. The small letters close to selected data points indicate respective bonds in the lower panels **b** and **c**. **b, c** Molecular structures of (**b**) investigated collagen crosslinks and (**c**) prevalent aminoacids in collagen. BDEs of bonds colored in black and orange (BDE <315 kJ mol⁻¹) were explicitly calculated. Residues R1, R1', R2 and R2' connect to the same collagen triple helix. The BDEs of the PYD bond denoted with the letter "a" was obtained after deprotonation of the pyridine-bound hydroxyl group. **d** Main electronic effects leading to the lowest found BDEs. The captodative effect describes the non-additive stabilization of the amino-carbonyl and carbonyl-amino group present at any $C_\alpha$ peptide radical. The lower part illustrates the increased radical delocalization upon deprotonation of PYD and DPD, lowering the BDE and RSE of bond "b" to that of bond "a".

We obtained bond dissociation energies (BDEs) of bonds from four major collagen I crosslinks (trivalent: PYD and DPD, divalent: HLKNL and deH-LNL), as well as of backbone bonds of collagen's main aminoacids (glycine, proline, and alanine). Further, we calculated radical stabilization energies (RSEs) of homolytic bond scission products to examine the origin of the large differences in BDEs, which can exceed 160 kJ mol⁻¹ across C-C bonds within a single crosslink. Figure 2a shows the BDEs of all calculated bonds against the summed RSE of the two radicals formed by each homolytic bond scission. All values can be found in Supplementary Tables 1–10. Overall, the linear correlation between low BDEs and large negative summed RSEs shows that BDE deviations from aliphatic C-C bonds can generally be explained by the stabilization of the formed radical centers by proximate chemical groups.

Across collagen crosslinks and backbone bonds, our results suggest that $C_\alpha$-$C_\beta$ bonds are promising bond rupture candidates (having BDEs ≤312 kJ mol⁻¹). The high stability (low RSE) of the formed $C_\alpha$ centered radical of −87 kJ mol⁻¹ explains the low $C_\alpha$-$C_\beta$ BDE, as do comparable literature values (−74.1[21] and −75.5 kJ mol⁻¹[5]). We attribute the comparably low RSE to the stabilizing effects of carbonyl-amino and amino-carbonyl substituents present in peptides, which are further increased when combined. Such non-linear donor/acceptor interactions are known as captodative effects (Fig. 2d)[22,23].

Interestingly, the presence of a second strong radical stabilizing system on the $C_\beta$ centered radical further lowers the BDE (Fig. 2, bond

"b") in the trivalent crosslinks PYD and DPD to just 282 kJ mol⁻¹. The stabilizing effect of the conjugated aromatic system becomes even more pronounced upon deprotonation of the pyridine-bound hydroxyl group, extending the resonant system and giving an exceptionally low BDE of 220 kJ mol⁻¹. Importantly, the pKa value of a reference compound (3-Hydroxypyridine methochloride, pKa = 4.96[24]) suggests a pKa of about 5, implying that PYD and DPD crosslinks are mostly deprotonated at neutral pH (and even more so their respective radicals).

We note that the electronic structures of radicals conjugated with the aromatic ring in PYD and DPD showed spin contamination, indicating multi-reference natures (see further tests and discussion in the Supplementary Information, in particular Supplementary Fig. 11). As the decreased BDEs of bond "a" and bond "b" appear plausible compared to literature RSEs and chemical intuition, we proceeded with these values but also varied PYD BDEs in the following analysis to assess the resulting uncertainty of rupture propensities in the fibril. Also note that throughout this study, we do not differentiate between low BDEs of PYD and DPD, as the calculation of four DPD bonds convinced us that all relevant BDEs are similar (details given in SI).

Focusing on divalent crosslinks HLKNL and deH-LNL, we found (in addition to $C_\alpha$-$C_\beta$ bonds) two more weak bonds in HLKNL and one more in deH-LNL. Again, the low BDEs can be understood by the strong radical stabilization effects of neighboring groups. For example, both

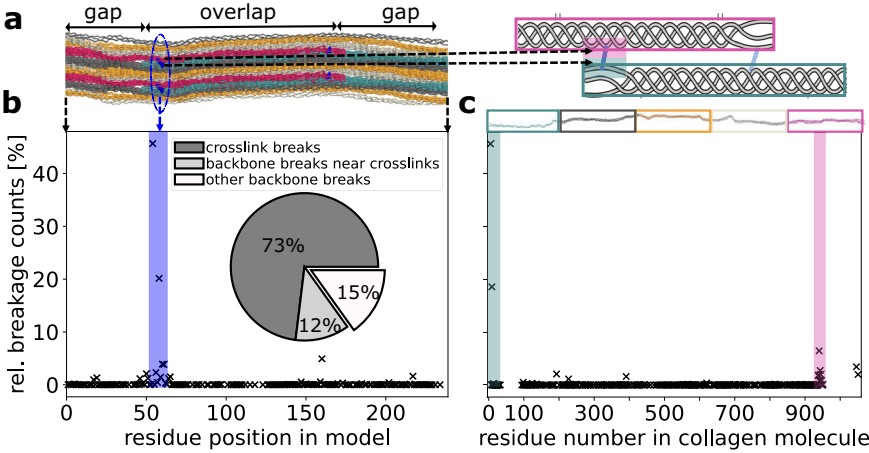

**Fig. 3 | Collagen ruptures primarily in crosslinks, but also unspecific ruptures occur everywhere.** Data were obtained from 63 independent MD simulations, including divalent and trivalent crosslink setups, with different collagen sequences and pulling conditions, as described in the methods section. For simulations with trivalent crosslinks, we used the deprotonated BDEs of PYD. In both panels **b** and **c** the same data is presented in two different reference systems. **a** Left side: Location of residues and crosslinks in our model along the fiber axis. Triple helices with the same phase are in the same color, respectively. Upper and lower residues participating in the same crosslinks (blue) appear in the same residue position in the model; see, for example, the blue encircled area. This view is used in panel **b**. Right side: Crosslinks along a whole 300 nm collagen molecule occur at the beginning (cyan area) and at the end of a triple helix (magenta area), shown as a cartoon with coloring as in the structure on the left. This reference system, counting along the collagen molecule, is used in panel **c**. **b** Propensity of bond breakages in our collagen model. Most ruptures concentrate in the crosslinked area, while there are some scattered backbone ruptures. Inset: Pie chart of summed-up ruptures in the crosslinks vs. the backbones in the crosslinked area (up to five residues before/behind) vs. elsewhere in the backbone. **c** Propensity of bond breakages within collagen fibrils. The crosslinked areas at the beginning and end of the molecules, as marked exemplary for the N-terminal crosslink in the shaded areas, are most prone to rupture. Note that both in **b** and **c** we only show data points where at least one rupture happened.

fragments of the weak $C_\delta$-$C_\epsilon$ bond of HLKNL have large RSEs of −68 and −70 kJ mol⁻¹ (literature: −65.3 kJ mol⁻¹ [25]). Furthermore, cleavage of the second weakest bond in HLKNL, $C_\epsilon$-$N_\zeta$, forms radicals stabilized captodatively by a carbonyl group in the beta position, as well as an N-centered radical with a moderate RSE of -30 kJ mol⁻¹ (literature: −25 kJ mol⁻¹ [5]).

Notably, none of the backbone bonds appear as likely rupture candidates. All calculated BDEs of backbone bonds, both $C_\alpha$-C and $C_\alpha$-N bonds, are considerably higher than those of the weak crosslink bonds (ΔBDE ≥24 kJ mol⁻¹). Within the backbone, $C_\alpha$-C bonds are overall weaker than $C_\alpha$-N bonds and are the primary rupture candidates of the protein backbone, in agreement with previous semi-empirical calculations[13], here largely independent of the amino acid type.

In essence, our quantum-chemical calculations identify bonds within crosslinks as weak bonds within collagen, in particular $C_\alpha$-$C_\beta$ bonds reaching values down to 220 kJ mol⁻¹ in the trivalent PYD crosslink. These can be backtraced to excellent stabilization of the resulting mechanoradicals.

### Ruptures occur dominantly in and around crosslinks, but also unspecific in backbones

Rupture propensity is determined not only by the thermodynamic strength of the bonds, as calculated above, but also by the actual force that is acting upon that bond. To directly predict relative rupture counts within a stretched collagen fibril, we utilized KIMMDY[12], a hybrid simulation scheme that invokes bond rupture in kinetic Monte Carlo steps during atomistic MD simulations. Bond rupture rates are calculated based on the bond's strength given by its BDE and its mechanical weakening by the force distribution through the protein in the MD simulation. For the simulations, we used atomistic collagen type I fibril models taken from ColBuilder[26] as described in the methods and shown in Fig. 1b. The simulated structures comprise one overlap and gap region in length and 41 triple helices in width (in the overlap region). We here focus on different enzymatic crosslinks, and our models do not incorporate so-called AGE crosslinks (advanced glycation endproducts). During the simulations at constant force, the system rearranges and reaches a new stretched (quasi-)equilibrium that mimics the conditions from previous experiments[2], in which homolytic bond ruptures were observed inside macroscopic collagen tendons at a sub-failure constant force regime. The individual extensions of the simulations are displayed in Supplementary Fig. 12 and show that, after an initial stretch, they quickly converge to strains in the range of 20–24%. With AGE crosslinks being absent, which are known to further stiffen collagen[27], the strain we reach appears plausible.

Notwithstanding the overall higher stability of the backbone bonds according to the quantum-chemical calculations, KIMMDY predicts that crosslink and backbone bond ruptures can both occur. Along the fibril structure (Fig. 3a, left), ruptures occurred randomly across the whole fiber axis (Fig. 3b), albeit at low relative counts. Most frequently, however, the crosslinks themselves break (at 73%). Also, the backbone in its vicinity has a strongly increased rupture propensity due to local stress concentration (12%). When mapping the rupture statistics onto the collagen I sequence (Fig. 3a, right), the prevalence of crosslink ruptures over widely distributed backbone ruptures again is evident (Fig. 3c): Only backbone regions where two crosslinked strands are overlapping and thus share the external stress, i.e., at the very beginning and end, are largely void of ruptures (areas without crosses in cyan and magenta boxes). Thus, the external force distributes such that some of the backbone bonds, in particular those in the vicinity of crosslinks, are weakened sufficiently to compete in terms of bond rupture rates with the intrinsically weaker crosslink bonds. In addition, backbone bonds are statistically still prone to rupture due to their sheer number, with up to -1500 aminoacids per crosslink in a fully crosslinked collagen I fibril. Consistent with a contemporaneous publication[13], we analyzed the rupture distribution along the GLY-X-Y pattern in collagen to be 0.7%-82.2%-17.0% in backbone bonds (excluding the common crosslink ruptures). As all backbone bonds have the same BDEs, this results from a stronger loading within bonds at the X-position.

The results shown in Fig. 3 are compiled from a set of KIMMDY simulations varying in crosslinks, collagen sequence (different species) and pulling conditions (see Methods). Importantly, in all of the

models and settings, we recover a competition between crosslink and backbone bond ruptures, i.e., a concentration of ruptures at and around crosslinks, and the random distribution of the remaining backbone ruptures along the sequence (Supplementary Figs. 6, 7). However, quantitative rupture counts can vary strongly and also depend on model choice and composition. For instance, if we assume acidic pH, that is, 100% protonated PYD, we have the lowest BDE of 282 kJ mol⁻¹ instead of 220 kJ mol⁻¹ (Fig. 2a, bond "b" versus "a"). In this case, the ratio shifts to 28%:41%:31% of breakages in crosslinks to those in adjacent backbone bonds to other unspecific backbone bonds, compare Supplementary Fig. 6c. Hence, as crosslinks now feature bond strengths closer to those of the backbone, ruptures throughout the backbone can compete more easily with those in crosslinks under the dynamic stress concentrations present in the stretched fibril. To conclude, crosslinks indeed represent weak spots, not only in terms of their BDEs but also in the context of the stress distribution across the collagen structure. The weaker the crosslinks are compared to the backbone, the more they can funnel initial ruptures into crosslinks and prevent breakages at other unspecific positions in the protein backbone.

## Trivalent crosslinks harbor sacrificial bonds

The trivalent crosslinks become more prevalent with age and connect two "arms", i.e., two modified lysine side chains of different strands from within one triple helix, to one lysine residue ("leg") of another triple helix (compare Fig. 2b). It has been speculated that due to this double connection, trivalent crosslinks provide more stability yet less elasticity. Having established crosslinks as major ruptures sites by our combined calculations, we next asked how the rupture propensity of the trivalent crosslink compares to its divalent counterpart (chemical structures shown in Fig. 2b), and how the redundant connectivity of the trivalent crosslink is compromised by rupture.

At first glance, the distribution of ruptures does not depend heavily on the crosslink type, with the exception of an even stronger concentration in the crosslinks for PYD. For the detailed rupture distributions, compare Supplementary Fig. 6b for PYD to Supplementary Fig. 13b for HLKNL. In Fig. 4a, however, we summed up all rupture rates within one simulation and grouped these total rates per crosslink type that we used in the model. The models with trivalent (PYD) crosslink exhibit clearly higher rupture rates. While this appears counter-intuitive at first glance, as the two arms can share the external mechanical load, the explanation is straightforward: the short arm of the trivalent crosslink has by far the lowest BDE due to the combination of the captodative effect and conjugation with the aromatic ring (compare Fig. 2d). While this leads to a higher rupture propensity for a first breakage, the overall crosslink remains intact due to the second arm. The low BDE of this bond also renders ruptures across the collagen fibril highly localized to the crosslink, with 80% ruptures occurring therein (Fig. 4a). Unspecific backbone ruptures halved compared to collagen with divalent crosslinks, from 20% to 10% of the rupture counts.

The rupture of a trivalent crosslink on one of the two arms leaves the linkage of the two triple helices overall intact. We next examined if subsequent ruptures occur primarily within the same crosslink or rather within other fully intact trivalent crosslinks. To this end, we continued nine pulling simulation of models with trivalent crosslinks after a rupture inside these arms, examined the new rupture rates, and compared to the situation of a fully intact pre-rupture fibril (Fig. 4b). Surprisingly, despite now lacking one force-bearing connection, the total bond rupture rate in the damaged trivalent crosslink decreases by ~3 orders of magnitude compared to the pre-rupture simulations. As bond strengths are unaffected, this decrease indicates that overall stress in this crosslink must have relaxed after its rupture. Indeed, we find an average distance increase of the crosslink $C_\alpha$ atoms after rupture by ~0.1 nm, as the longer arm can

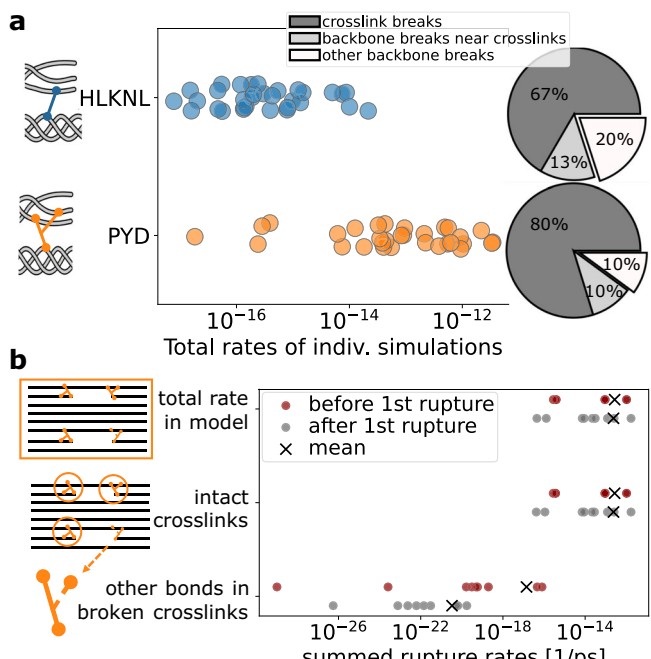

**Fig. 4 | Trivalent crosslinks break faster, but (at first) without loosing connection. a** Comparison of total primary bond rupture rates in simulations with models having divalent (HLKNL) vs. models having trivalent (PYD) crosslinks, which show overall higher rates. The ratio of crosslink ruptures to other breakages confirms that the increased rupture rates are indeed due to higher propensities in the crosslinks. **b** Comparison of secondary bond rupture rates before and after the first arm in one PYD crosslink broke. The rate in the second arm even decreases, being several orders of magnitude lower compared to the sum in the remaining model. Hence, other bonds will break first and the fibril stays overall intact. Each data point is from one of nine simulations with trivalent crosslinks, which we continued after the first rupture in the short PYD arm.

now elongate and use the previously hidden length (Supplementary Fig. 8).

Also, the remaining intact crosslinks that still each harbor the extraordinarily weak bond dominate the total of all rates in the simulation. Another rupture in the damaged crosslink is highly unlikely, and this crosslink stays intact. Instead, subsequent ruptures are predicted to mainly occur in the same type of weak $C_\alpha$-$C_\beta$ bonds in other triple helices. Our combined data in Fig. 4 suggests that the PYD and DPD crosslinks are less stable with respect to first ruptures, without compromising fibril integrity.

## Experiments confirm covalent backbone rupture and altered crosslinking

We next set out to experimentally validate our predictions of both crosslink and backbone covalent bond ruptures in collagen. Electron-paramagnetic resonance (EPR) spectroscopy is a highly sensitive method for the detection of mechanoradicals, and has been used for this purpose successfully[2]; however, it does not allow spatial resolution, and in addition, rapid migration onto DOPA prohibits the primary rupture site identification. We instead resorted to gel electrophoresis combined with mass spectrometry. Figure 5a shows the protein molecular weight distribution of solubilized collagen polypeptides from stressed and unstressed (control) collagen rat tail tendon resolved by an SDS-PAGE gel. We recovered the expected band pattern, including the monomeric $\alpha$-chains ($\alpha1(I)$ and $\alpha2(I)$), as well as higher-order oligomers (e.g. $\beta$-chains, $\gamma$-chains and $\delta$-chains). For the stressed samples, we additionally observe bands in the mass regime of fragmented chains, indicating backbone rupture of single collagen strands. To quantify the proteolytic breakdown in stressed and control

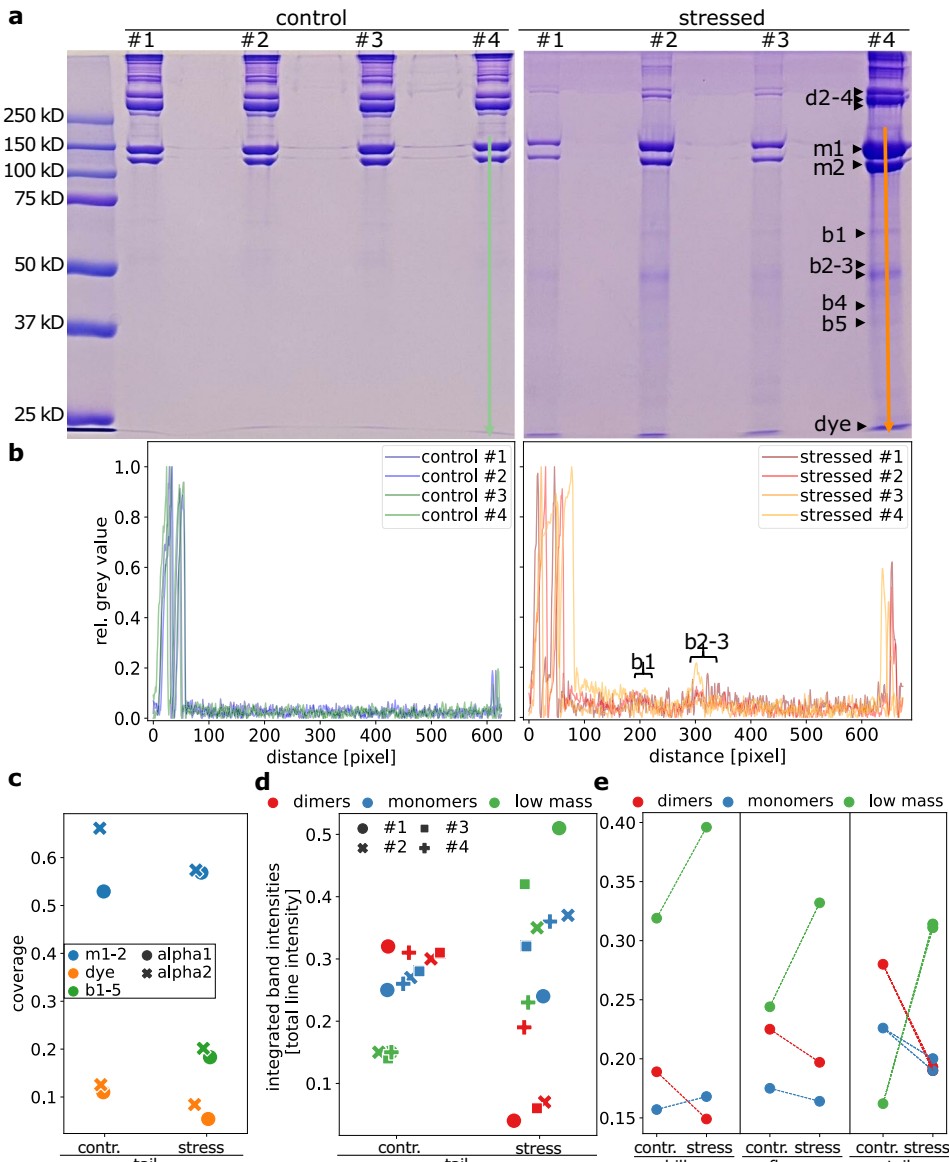

**Fig. 5 | Experiments support the simulations of covalent backbone and crosslink rupture in stressed tendon collagen. a** SDS-PAGE analysis reveals altered patterns of bands and their intensities between stressed and control tendon collagen. The four lanes are technical repeats, the experiment was repeated independently for a second biological sample set, see Supplementary Fig. 10. We indicate lines used for analysis in panel **b** in lane #4. The bands b1–5, cut out for the mass spectrometry, are marked with arrows. **b** Densitometry to quantify stained collagen α-chains and degradation products on SDS-PAGE. We show the normalized inverted gray value along the lanes. **c** Percent sequence coverage of collagen α1(I)-

and α2(I)-chains from the bands analyzed by mass spectrometry. Collagen fragments are present in the mass regime of monomers m1 and m2, low mass bands b1–b5 (averaged, individual data in Supplementary Fig. 9) and even the dye front. **d** Integrated intensity in the two monomeric bands (blue) and three dimer (red) bands, as well as the total area below the monomers (incl. dye front, green), normalized for each lane with the total intensity. **e** Same analysis as in panel d for additional gels provided in Supplementary Fig. 10 from Rat tail, flexor, and achilles tendons.

---

tissues, we obtain the densitometry of stained collagen chains and peptide fragments across samples with ImageJ[28] in Fig. 5b. Stressing the samples gives rise to fragments of masses lower than those of full-length collagen chains, across a wide range of masses down to small fragments running at the dye front.

Second, we identified intact α-chains and lower molecular weight polypeptides by mass spectrometry. We took them from regions as indicated in Fig. 5a from stressed samples #2 and #4, because peptide fragments from samples #1 and #3 were below the limits of detection. The sequence coverage of the collagen α-chains, obtained with Proteome Discoverer and displayed in Fig. 5c, confirms the presence of both α1- and α2- chain fragments in all these areas, even in the lowest mass fractions of the dye front (see Supplementary Fig. 9 for coverage

across the collagen sequence). Overall, this experimental analysis supports the unspecific breakages found in the simulations (compare Fig. 3c), as the stressed samples contained multiple α-chains peptide fragments indicative of unspecific backbone breakages in the fiber. Given that the intensity of low molecular weight bands increases upon mechanical stress, we attribute this indeed to mechanical loading.

Lastly, we used variations in the collagen protein mass patterns and their intensities as observed by SDS-PAGE as a qualitative measure of changes in collagen crosslinking. Here, we test our hypothesis of loss of connection between the molecules by rupture of crosslinking upon mechanical stress, as predicted by our simulations (Fig. 3b). Thus, we expect to observe less oligomers (β-, γ-, and δ-chains) and instead more monomers. Indeed, there is precedent for

predicting alterations in collagen crosslinking through changes in patterns of bands in SDS-PAGE gels, whether caused by chemical treatment or genetic disorder[29–31]. It is apparent in Fig. 5a that the concentration of oligomers is decreased in the stressed samples. We quantify the integrated intensities of dimers and monomers in each lane in Fig. 5d, after normalizing to the overall intensity to account for the different signal strengths in the lanes. In the control, dimers account for slightly more of the total intensity in their lane than monomers. In contrast, after mechanical treatment, dimer intensity decreases, and the monomeric $\alpha$-chains make up a higher fraction in the stressed samples. Hence, stress on the tissue disrupted the crosslinks typically found in these higher molecular weight collagen polypeptides. Again, this is consistent with the crosslink breakages predicted from our simulations.

Together, the experimental evidence of both backbone rupture and altered crosslinking supports the detailed simulation results. The increase in lower molecular weight fractions is indicative of unspecific backbone ruptures, and the shift from oligomeric combinations to monomers can be explained by ruptures of crosslinks that otherwise held those oligomers together. In the simulations, the exact ratio of backbone versus crosslink rupture depended on parameter details as pointed out above, but that competition was always recovered. Similarly, which of the two mechanisms are more prevalent in the experiments remains yet to be clarified in future work and likely depends, among other factors, on species and tissue age.

## Discussion

Following the hierarchical structure of collagen, we identified collagen type I breakage sites by means of a scale-bridging approach. High-level quantum-chemical calculations returned bond dissociation energies that we incorporated into KIMMDY, yielding rupture statistics, which in turn were validated by experiments. The KIMMDY approach is more efficient than hybrid quantum mechanical/molecular mechanical (QM/MM) techniques[32,33], especially given that the initial rupture sites are unknown and the whole molecule can potentially react. Further, having now detailed BDEs at hand, it is also more accurate than reactive force fields[34], which would also not scale well to these large atomistic models. KIMMDY operates in the (macroscopic) sub-failure regime of collagen with occasional internal bond ruptures, which would otherwise not be computationally accessible. This regime is consistent with previous experiments showing radical generation for time scales of seconds to hours[2], and can, for instance, be seen as a common scenario of heavy tendon loading without overall fiber rupture.

Albeit more efficient than QM/MM, KIMMDY is still limited to sub-micrometer-scale atomistic systems. Our simulations thus can not take structural features beyond this length scale into account, such as heterogeneities in crosslinking or flaws that can aggravate stress concentrations[2]. We reach most of the strain already after 10–20 ns and assume a separation of time scales with (un-)loading times beyond the rupture kinetics. We did not consider fast non-equilibrium processes like a catastrophic fiber failure, which could be a possible extension of KIMMDY for future work. A particular strength of KIMMDY is, however, that it takes structural rearrangements as well as effects of internal friction on the distribution of forces through the structure and, thus, on the resulting rupture kinetics directly into account. It, therefore, is well suited to study ruptures in stretched collagen, which has previously been shown to feature conformational changes on various structural levels, from proline ring flips to helix rearrangements and beyond[13,35–37].

We observe the rupture propensity of covalent bonds in collagen to be determined by a competition between how weak a bond intrinsically is, that is, low BDEs, and how much force acts on this bond, that is, the stress distribution in the fibril. We find this interplay to depend on various conditions, most importantly the crosslink type (and BDEs

therein) and the external force application, but also on computational details in KIMMDY as described in the methods section. We have limited the computational part of this study to the most common collagen type I, but find decisive features like crosslink sites and DOPA positions to be highly conserved in other minor tendon collagen types III and V (see multiple sequence alignment in Supplementary Fig. 14). Larger and, for this reason, potentially coarser models will be needed to more exhaustively sample other crosslinks, species and mixtures of crosslink types at varying mol/mol, as well as the impact of higher-order structural features of collagen. However, notwithstanding the sensitivity of quantitative results with regard to parameters, a robust picture has emerged: Crosslinks harbor the weakest bonds. In particular, the captodative effect stabilizing $C_\alpha$-centered radicals, and in trivalent crosslinks, the additional resonance with the aromatic ring gives rise to exceptionally low BDEs. Hence, they break most often. However, the stressed protein backbone still breaks randomly as well due to stress localization, but also simply due to the vast amount of bonds under force. The lower the difference between the backbone and crosslink BDE, the higher the share of backbone ruptures in that competition. We could also observe both backbone ruptures and altered crosslinking in our SDS-PAGE gels. The picture emerging from the experiments with different Rat tendons (tail, flexor, achilles) corroborates the simulation results. However, we also note that future computational and experimental work is needed for a more quantitative confirmation of the predictions.

Protein $C_\alpha$ radical stabilization by the captodative effect was found before, e.g. when inducing radical formation via UV light[38], or by mass spectrometry experiments[39]. These studies support our result, and we here extend the decisive role of the captodative effect in proteins to mechanical bond rupture and mechanoradical formation in a stretched protein material.

The stable methylpyridine radical in trivalent crosslinks extends the set of aromatic moieties known to stabilize radicals in proteins, namely histidine, tyrosine, tryptophan, and DOPA, as e.g. in ribonucleotide reductase[40–42], or collagen[2]. The benzylic methylpyridine radical is chemically remarkably similar to Vitamin B6, with well-known radical-scavenging properties[43,44]. A DOPA radical, however, is even more stable, with a BDE for hydrogen abstraction (X-H) as low as 260 kJ mol$^{-1}$ for a DOPA anion[45]. We here determined a BDE(X-H) of around 337 kJ mol$^{-1}$ for the deprotonated PYD; hence, in the context of the collagen structure, radical migration to DOPA seems likely from a thermodynamic viewpoint. A recent study experimentally confirmed the presence of DOPA, often nearby crosslinks, in different collagenous tissues and its radical-scavenging properties[19].

Our data propose several advantages of the unique trivalent crosslinks of collagen. First, its rupture results in exceptionally stable radical species, reducing unspecific reactions such as uncontrolled migration. The stabilizing effect is enhanced by the localization of breakages, as low crosslink BDEs funnel ruptures to sites where a rapid transfer to nearby DOPAs is likely. Thus, further damage to the material is mitigated. Secondly, we find the mechanical strength of the trivalent crosslink to be pH-sensitive, i.e., BDEs depend on the protonation state of PYD (Fig. 2). We speculate that collagen's rupture propensity could adapt to variations in physiological pH, e.g., in muscle after exercising[46,47] or under inflammatory conditions[48].

Thirdly, we find the weak bonds within the trivalent crosslinks to act as sacrificial covalent bonds. Stress relaxes in the trivalent crosslink after the initial rupture, and subsequent ruptures occur in other crosslinks, according to our computational analyses. Mechanically, the sacrificial arm in the trivalent crosslink can increase the toughness of the material due to energy dissipation and relaxation afterward, similar to what is known from sacrificial bonds in polymers[17]. Finally, radicals formed in crosslinks can migrate rapidly to nearby DOPAs and then, in the presence of water, react with superoxide radicals to create $H_2O_2$, as has previously been shown in pulling experiments on collagen fibers[2].

In this way, the breakage of one of the two arms offers a possibility to already start signaling high loads in collagen, triggering repair mechanisms and other reactions, even before macroscopic failure occurs. Speculating further, a self-healing effect similar to what can be observed in polymers[49] or biological fibers[50] could also come about by either direct DOPA recombination or by the formation of DOPA-Lys and DOPA-His crosslinks[51,52].

Collagen's remarkable trivalent crosslink chemistry appears perfectly adapted towards handling mechanical stress and mechanoradicals and, as such, can inspire the design of synthetic (biocompatible) material systems with high toughness, self-healing, or self-reporting strategies. Our results propose that collagen might have evolved into a high-performance biomaterial by means of highly specific mechanochemical routes that enable collagen to self-report on rupture while preventing further unspecific damage. If this specific mechanochemistry is similarly at play in other biomaterials such as elastin, and to what extent it plays a role in tissue ageing, remains to be elucidated.

## Methods

### Quantum-chemical calculations

Molecular structures were built in GaussView. Initial phi and psi values of aminoacids were set to common values for the respective amino acid in a Ramachandran plot, based on a stretched collagen MD simulation. All radical structures were built from already geometry-optimized molecular structures. H-capped radicals, used for radical stabilization energy calculations, were built by hydrogen addition to geometry-optimized radicals. All structures were geometry optimized at the BMK/6-31+G(2df,p) level of theory in Gaussian09, following a slightly modified G4(MP2)-6X protocol described elsewhere[20]. Before the calculation of G4(MP2)-6X energies, structures were assured to have converged and be free of imaginary frequencies. More detailed information can be found in the Supplementary Information.

### BDE and RSE calculations

Bond dissociation energies (BDEs) were calculated as described in Equation (1) based on G4(MP2)-6X enthalpies ($\Delta H_{298K}$) of a given molecule ($R_1$-$R_2$) and its fragments formed after homolytic bond scission ($R_1^*$, $R_2^*$).

$$BDE = \Delta H_{298K}(R_1^*) + \Delta H_{298K}(R_2^*) - \Delta H_{298K}(R_1-R_2) \quad (1)$$

Radical stabilization energies (RSEs) show how well a radical is stabilized relative to the reference methyl or ammonia radical. They were calculated with BDEs of $CH_3$-H bonds for C-centered and $NH_2$-H bonds for N-centered radicals. G4(MP2)-6X enthalpies of radicals ($\Delta H_{298K}(R_1^*)$, $\Delta H_{298K}(Ref^*)$) and their H-capped counterparts ($\Delta H_{298K}(R_1-H)$, $\Delta H_{298K}(Ref-H)$) were used to obtain RSEs as given in Equation (2).

$$\begin{aligned} RSE(R_1^*) &= BDE(R_1-H) - BDE(Ref-H) \\ &= \Delta H_{298K}(R_1^*) - \Delta H_{298K}(R_1-H) - \Delta H_{298K}(Ref^*) + \Delta H_{298K}(Ref-H) \end{aligned} \quad (2)$$

### Molecular dynamics simulations

All MD simulations were carried out with the software package GROMACS, version 2018 or 2020[53], using the amber99sb-star-ildnp force field[54,55]. We employed 2 fs time steps with LINCS[56] constraints on h-bonds. In order to account for the asymmetric stretch of bonds under these high mechanical loads and also in order to enable bond scissions, we used Morse potentials for the bond interactions. Simulations were conducted at 300 K and 1 bar, which were maintained by a v-rescale thermostat[57] and a Parrinello–Rahman pressure coupling[58], respectively. We cut off both Lennard-Jonas and Coulomb interactions at 1.0 nm. The different collagen models were solvated in

TIP3P water leading to system sizes of ~2.5–2.9 million atoms, for which periodic boundary conditions were chosen. After energy minimizations, we conducted equilibrations in NVT and NPT ensemble for 10 ns each.

For production, we subjected the collagen fibrils to a constant force of 1 nN per chain in four different schemes: One is that we distributed the forces equally, leading to 3 nN on each triple helix, pulling from both sides. In schemes two and three, we also pulled equally on all strands, but just from one side and restrained the other side of the molecule. We used this as a control that our results are independent of the time the force needs to propagate through the system and of internal friction. In the last scheme, we inhomogenously distributed forces on the inner triple helices by drawing forces from a Gaussian distribution with the same average of $F_{av} = 1$ nN and width of $\sigma = F_{av}/3$. The ring of outer helices was still subject to 1 nN to avoid sliding the chains against each other. We refer to this scheme as shear pulling, it mimics more imperfect biological samples. In all cases, we additionally employed torque restraints to prevent the unwinding of the capped collagen triple helices[59]. The applied stress level will lead to a new, stretched equilibrium of the fibril that corresponds macroscopically to a sub-failure regime that leads to occasional bond rupture, consistent with the scenario of radical generation in stressed tendons from previous work[2]. The individual extensions of the simulations are displayed in Supplementary Fig. 12, and show that after an initial stretch, we quickly converge to constant strains in the range of 20–24%.

Overall, we used collagen fibril models from three different species *Rattus Norvegicus*, *Pongo Abelii*, and *Loxodonta Africana* (as published on the webserver ColBuilder[26]), which yet again varied in their crosslink position and also crosslink type (divalent HLKNL or trivalent PYD). Including replica, we, in total, conducted 63 simulations. In each instance, the collagen fibrils were pulled for 100 ns before executing the KIMMDY scheme of bond rupture, as discussed below. For nine simulations with PYD crosslinks, we thereafter conducted a second cycle of 100 ns pulling to investigate secondary breakages. All models and their replica make up more than 70 MD simulations with over 100 ns each in this manuscript.

### Bond breakages in KIMMDY

We use our previously developed simulation scheme **Ki**netic **M**onte **C**arlo/**M**olecular **Dy**namics (KIMMDY)[12] to enable bond scissions on the molecular level. In short, it takes the bond elongations from the MD simulations as input for a Bell-type model to calculate rupture rates. Bonds are modeled with Morse potentials, of which the widths and equilibrium bond lengths are taken from the regular force field parameters. The initial barrier height given by the BDEs (from the quantum calculations) is effectively lowered by the work that acts on the individual bond, leading to a new, effective barrier that depends also on the force in the bond (i.e., on its average elongation in the MD). The individual bond rupture rates are then obtained from the barrier crossing problem of this new effective potential and are used in a Kinetic Monte Carlo step determining the scission site and the associated propagation of the system.

After the aforementioned 100 ns pulling MD simulations, 5000 bond elongations per breakage candidate, taken from another 1 ns of simulation now using a PLUMED-patched[60] Gromacs version, were averaged to generate converged bond rupture rates as shown previously[12]. For each of these ensembles of rates, the Kinetic Monte Carlo step was invoked 10,000 times to sample the rupture propensities. Taken together, this procedure ensures that both the rates and the resulting breakage distribution are converged to a degree where the statistical fluctuations within simulations are negligible compared to the differences in replica. In the cases where a second rupture was simulated, which we did for nine simulations with PYD crosslink, we adjusted the topology and relaxed the system with a smaller 0.2 fs time step before pulling for another 100 ns and repeating the procedure.

In contrast to the original implementation, which used default backbone values for BDEs and did not differentiate between the various crosslink chemistries, we here utilize the BDEs that we calculated in this manuscript as barrier heights. Concerning peptide backbone bonds, we found that $C_\alpha$-N BDEs of alanine and glycine are identical within methodological errors. The same was the case for $C_\alpha$-C bonds of alanine, proline, and glycine, even if we would expect some difference due to different chemical environments given by an amino acid's side chain. For subsequent simulations, we used the resulting averages of 379 and 341 kJ mol$^{-1}$ for all $C_\alpha$-N and $C_\alpha$-C bonds, respectively. Note that this is different from our previous work, where we assumed that $C_\alpha$-N and $C_\alpha$-C bonds have similar BDEs. Hence, beside the collagen-specific crosslinks, our method KIMMDY also improves generally for other protein use cases with these new backbone values.

Secondly, we also revised the way KIMMDY calculates forces. We noticed that rupture probabilities varied depending on the amino acid type. While these differences did not qualitatively change our results, we carried out a detailed investigation on this, as can be found in the supplement, see in particular Supplementary Fig. 13. In summary, we attribute this effect partially to influences of the force field and partially to actual differences in the systems, which can hardly be separated due to the nature of the KIMMDY protocol.

## Experiments

For the first experimental data set, Rat (*Rattus Norvegicus*) tails were sampled earlier from female rats, 5–6 months old, snap-frozen, and stored at −80 °C until the day of preparation. Tendon samples from the tail were excised by pulling out single collagen fibers. The fibers were collected into ice-cold Dulbecco's phosphate-buffered saline (DPBS) w/o calcium and magnesium. Collected samples were then briefly washed by over-head agitation at 4 °C and collected by centrifugation ($500 \times g$ force, 2 min, 4 °C). This step was repeated twice. After washing, the fibers were equilibrated in the atmosphere for 1 h. Dried fibers were crushed in a porcelain mortar, with a manually applied stress level that is likely above the simulation setup. In previous EPR experiments[2], this led to a signal increase without a change in signal type. Next, they cooled and filled with liquid nitrogen, for 5 min. Control samples were left untreated. Both crushed and control samples were collected on dry ice and stored at −80 °C until shipment.

For the second experimental set, tail tissue was sampled and prepared as described above. Achilles' and flexor tendons were dissected from hint legs and washed in the same manner. All washed tendon samples were freeze-dried for 48 h. Dried tendons were split randomly into a control and cryo-milling group, then frozen at −80 °C until the day of treatment. We switched from crushing to milling as it is more quantitatively controllable. Cryo-milling in liquid nitrogen flow was conducted on a Retsch Cyromill (Haan, Germany) with a 25 mL zirconium oxide chamber and a 15 mm marble for $2 \times 2.5$ min at 30 Hz. The control group was left untreated. Control and milled samples were then shipped for further analysis.

After shipment, intact type I collagen was solubilized from the lyophilized rat tissue by SDS extraction in SDS sample buffer for 5 min at 100 °C. Sample loads were normalized to the dry weight of the tendon in the first data set. As sample uptake in stressed material varied, loads were normalized manually to the intensity of the $\alpha$-chains in the second data set. SDS-extracted collagen $\alpha$-chains and collagen peptides were resolved by SDS-PAGE and stained with Coomassie Blue R-250. Collagen peptides and $\alpha$-chains were cut from SDS-PAGE gels and subjected to in-gel trypsin digestion. Electrospray mass spectrometry was carried out on the trypsin-digested peptides using an LTQ XL linear quadrupole ion-trap mass spectrometer equipped with in-line Accela 1250 liquid chromatography and automated sample injection[61,62]. Thermo Xcalibur software and Proteome Discoverer software (Thermo Fisher Scientific) were used for peptide identification and sequence coverage analysis. Tryptic peptides were also identified manually by calculating the possible MS/MS ions and matching these to the actual MS/MS spectrum. Protein sequences used for MS analysis were obtained from the Ensembl genome database.

Densitometry of stained collagen chains (($\alpha$, $\beta$, and $\gamma$) from SDS-PAGE was determined using NIH ImageJ software[28]. Briefly, using ImageJ, the densitometry of each SDS-PAGE lane was calculated from a 32-bit gray image of the SDS-PAGE after the background (measured as the mean intensity of a line between lanes) was subtracted.

### Reporting summary

Further information on research design is available in the Nature Portfolio Reporting Summary linked to this article.

## Data availability

The QM data generated in this study (including data for Fig. 1) is provided in the supplementary tables. Run input files (enabling reproduction) for MD and KIMMDY simulations used in this study, derived breakage counts per simulation (including data for Figs. 2, 3, 4) and experimental data (for Fig. 5) and uncropped pictures of gels are all available in a heiDATA repository[63]: https://doi.org/10.11588/data/HJ6SVM. Full raw MD simulation data is too large to deposit and available on request. Amino acid sequences for Col1a1 and Col1a2 were obtained from the Ensembl database: https://useast.ensembl.org/Rattus_norvegicus/Info/Index.

## Code availability

KIMMDY is available as already published previously[12]: https://github.com/hits-mbm/kimmdy.

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

## Acknowledgements

This work was supported by the Klaus Tschira Foundation [F.G. and G.G.]. This research was conducted within the Max Planck School Matter to Life supported by the German Federal Ministry of Education and Research (BMBF) in collaboration with the Max Planck Society [B.R., C.K., A.Ü., and F.G.]. This project has also received funding from the European Research Council (ERC) under the European Union's Horizon 2020 research and innovation program (grant agreement No. 101002812) [F.G.]. The authors gratefully acknowledge the Gauss Centre for Supercomputing e.V. (www.gauss-centre.eu) for funding this project by providing computing time on the GCS Supercomputer SuperMUC-NG at Leibniz Supercomputing Centre (www.lrz.de) [B.R. and F.G.]. We further acknowledge funding through the Deutsche Forschungsgemeinschaft (DFG, German Research Foundation) under Germany's Excellence Strategy - 2082/1 - 390761711 [F.G.]. For the publication fee we acknowledge financial support by Deutsche Forschungsgemeinschaft within the funding programme "Open Access Publikationskosten" as well as by Heidelberg University. We thank Marilyn Archer for technical assistance with the collagen extraction and SDS-PAGE. We thank Helmut Grubmüller and the respective department at the Max Planck-Institute for Multidisciplinary Sciences for useful discussions and suggestions about the research in this manuscript.

## Author contributions

B.R. conducted together with A.Ü. MD simulations, together with C.K. KIMMDY simulations, and analyzed the simulation data and the experimental results. C.K. conducted and analyzed QM calculations. M.K. prepared samples for experiments and supported their interpretation. D.M.H. conducted SDS-page gels and mass spectrometry experiments and supported their interpretation. D.M. conducted multiple sequence alignments of collagen types. K.R. and G.G. supervised QM calculations and supported their interpretation. F.G. supervised the project. B.R., C.K., and F.G. wrote the manuscript. All authors commented on the manuscript and contributed to it.

## Funding

## Competing interests

The authors declare no competing interests.
