## [Peer Review File · Nature Communications]

Collagen breaks at weak sacrificial bonds taming its mechanoradicalsReviewers' Comments:

Reviewer #1:

Remarks to the Author:

The authors have investigated the breaking of chemical bonds in collagen as a response to mechanical forces. The authors combine multiple computational and experimental approaches. Their key finding is that the bonds in and close by crosslinks are most likely to rupture, and the authors relate this to the weakness of the bonds and the stabilisation of the resulting radical species.

While the manuscript contains some interesting new findings, a number of points need to be addressed before any possible publication. In my review, I will focus on the computational part of this study.

1. The fact that crosslinks are very likely rupture sites has been described before, and supporting work is cited. Although the claim that this is mainly based on previous work by the authors is misleading. This might be due to unfortunate phrasing, but this part of the introduction certainly needs attention, giving appropriate credit to references [11] and [12]. This point extends to reference [13], which also describes bond failures around a crosslink, and further highlights the importance of the backbone arrangement in bond breaking locations.

2. Later in the introduction the claim is made that collagen has evolved to 'self-report on rupture while preventing further unspecific damage'. There is no further discussion of this hypothesis nor is any evidence provided in support. I agree that the findings hint at a well-designed system, but that this is due to evolution needs to be evidenced.

3. The calculations of bonding energies are interesting, but require context. If we consider a purely thermodynamic event and look at equilibrium probabilities, the reported energies are key. However, bond ruptures due to mechanical forces are likely a non-equilibrium event.

4. It has been reported that collagen backbone arrangements change significantly with mechanical forces and so do the corresponding configurations of amino acids in collagen. Some of this work is cited by the authors, though not in this context. How are such effects accounted for in this work? It seems to me that the authors explicitly exclude internal friction and uneven distribution of forces, although this is likely what would be encountered in collagen tissues.

5. Related to these points is the question about instantaneous bond rupture vs. slow loading. Intuitively, I would assume the former is more likely, yet the authors seem to look more at the later, as they consider slow loading, distribution of forces across the fibre etc. What are the reasons for this setup?

6. The authors discuss the impact of different amino acids somewhat, but then conclude that they don't actually matter despite revising their model to account for different amino acids. However, this seems to rely on the fact that the bond energy differences are very small, and this is what is used in the KMC simulations. However, this again does not account for potential alternative configurations or non-equilibrium effects. It is also in contradiction to the preference in of force accumulation in specific bonds in the back bone, partly due to the role played by Pro residues, as found in the semi-empirical calculations cited by the authors.

7. How do the authors test convergence in their KMC simulations?

8. Line 348 - the total accumulated simulation time is meaningless. Total simulation time has no relation to convergence or sampling and should not be used.

Reviewer #2:

Remarks to the Author:

This is an interesting paper providing novel detailed analysis on the mechanics of collagen, and bond rupture propensity therein. While the calculations and comparison of different bond strengths is novel, this reviewer has several concerns about suggested multiscale mechanisms that govern bond rupture in mature fibrils.

MD simulations provide a basis for KIMMDY simulations, however, the equilibration times before pulling simulations are very short (10 ns).

The authors discuss that the multiscale topology of collagen determines how force is distributed throughout the structure and thereby modulates bond rupture. Based on this supposition, how is the location of crosslinks identified in the models? Is the selected location of the crosslinks consistent with the heterogeneity expected? What is the control for crosslink location? Do crosslinks differ across species and how is this captured in simulation? Furthermore, experiments are limited to a single species – can this choice be explained?

Authors discuss primary and secondary rupture, in particular in the context of trivalent crosslinks. They explain that “the total bond rupture rate in the damaged trivalent crosslink decreases by ~3 orders of magnitude compared to the pre-rupture simulations, indicating that stress in this crosslink overall relaxed after its rupture.” Can the authors measure this stress distribution directly to support their claim of the mechanism? There appear to be different contributing mechanisms to rupture – bond strength and force distribution in the vicinity of the bond based on the topology of the bond location. The rupture rate does not directly explain which mechanism is responsible.

The experimental results support both breaking of backbone and crosslink bonds, but does not compare their breaking propensity. As a result, it is not clear that the experiment confirms the trends observed in the models.

Can the authors specify in their methods (or SI) further details about the parameterization within KIMMDY from calculations? Specifically, for multi-armed crosslinks, the mechanical response may differ along different directions; is this captured in the KIMMDY scheme?

Reviewer #3:

Remarks to the Author:

This exciting manuscript provides a highly interesting and original perspective on the question how the hierarchical structure of collagen contributes to the remarkable mechanical properties of this protein. Traditionally this problem has been thought of purely in mechanical terms, but this work demonstrates that in addition there is a unique mechanochemical mechanism at work. By advanced scale-bridging simulations, the authors convincingly show that the structure of collagen is set up such that trivalent crosslinks can serve as sacrificial bonds and also such that mechanoradicals created by covalent rupture are quickly stabilised. The simulation predictions are qualitatively supported by experiments where SDS-PAGE and mass spectrometry is used to analyse the fragments liberated by rat tail tendon upon mechanical stress. The analysis of these experiments should be clarified. Also, the authors should clarify the generality of their findings for different collagenous tissues and animal species.

Comments

1. The experiments in Fig 4 show corroborating evidence that is consistent with the simulations, but do not provide a true, stringent test of the predictions. A key prediction (Fig 4) is for instance the key role of trivalent crosslinks, which the authors could test experimentally by either comparing rupture patterns (i.e. fragment fingerprints) for collagenous tissues that differ in trivalent crosslink content (e.g. because of different age), or by comparing rat tail tendon against in vitro-assembled collagen

fibrils, which only have divalent crosslinks.

2. Regarding the experiments:

a. the authors should clarify what is the nature and magnitude of the mechanical stress, which is unclear from the described procedure (dried fibers were crushed, cooled and filled with liquid nitrogen?), and whether or not this is representative of the effects of mechanical stretching they explore in the simulations.

b. The authors use rat tail tendon. What is the age and crosslinking pattern for this tissue? Typically tendon has more immature crosslinks than other tissues such as dermal tissue.

c. Why are there collagen fragments found for unstressed collagen?

d. Why is there strong variability for the 4 samples in Fig 5a (right)?

e. The authors should improve the annotation of the data in fig 5. It is unclear from the figure which bands from (a) are seen in (b) (arrows could be used) and which bands are analysed (separately or pooled?) in c and d.

3. Data for MD simulations with 3 different collagen sequences are pooled together in Fig. 3. Why were the data pooled? And did the analysis for different collagen sequences show any differences among the 3 species?

4. For context, it would be helpful if the authors could discuss both the stress and corresponding strain in their simulations and how it compares to typical stresses and strains encountered in collagenous tissues.

5. The rupture simulations are performed on atomistic collagen type I fibril models. Can the authors speculate how higher-order fiber structures, which depending on tissue type can be twisted or not, will affect their mechanism? And collagen fibrils in tissues are made of mixtures of collagen I and minor fibrillar collagen types (e.g. III, V), how will this affect rupture?

Minor comments

1. The authors should change the terminology by which they describe the SDS-PAGE gels; "banding pattern" in the context of collagen typically refers to collagen's axially ordered D-banded fibril structure.

2. The authors should clarify their statement at the bottom of p2 "SDS-PAGE coupled to MS supports altered collagen crosslinking" (altered as compared to what?)

3. The authors should clarify their statement on p7 "The weaker the crosslinks are compared to the backbone, the more they can prevent any other unspecific position in the protein backbone."

4. The authors should clarify what they mean by "as a measure of collagen crosslinking quality" on p10.

REVIEWER COMMENTS

We thank all reviewers for their time, the discussion points and their remarks that have contributed to a – as we believe – now clearly improved version of the manuscript.

We added a new set of experiments that generalizes the observation of collagen rupture to different tissues (new panel Fig. 5e) and several new Supplementary Figures, most importantly with additional analyses of convergence of the simulations, and a sequence alignment across collagen types. We clarified the methodological approach and added relevant references. Further, according to Nat. Comms. policy, we have changed Fig.4b) to not only show averages as before, but all individual data points, because of a sample size $n < 10$. Additionally to the revised manuscript and SI, we added a source data folder and a data and code availability statement.

Reviewer #1 (Remarks to the Author):

The authors have investigated the breaking of chemical bonds in collagen as a response to mechanical forces. The authors combine multiple computational and experimental approaches. Their key finding is that the bonds in and close by crosslinks are most likely to rupture, and the authors relate this to the weakness of the bonds and the stabilisation of the resulting radical species.

While the manuscript contains some interesting new findings, a number of points need to be addressed before any possible publication. In my review, I will focus on the computational part of this study.

We thank the reviewer for the overall positive evaluation and will address the points one-by-one in the following.

1. The fact that crosslinks are very likely rupture sites has been described before, and supporting work is cited. Although the claim that this is mainly based on previous work by the authors is misleading. This might be due to unfortunate phrasing, but this part of the introduction certainly needs attention, giving appropriate credit to references [11] and [12]. This point extends to reference [13], which also describes bond failures around a crosslink, and further highlights the importance of the backbone arrangement in bond breaking locations.

We agree that the phrasing was not optimal and have re-written it to point out that all the cited studies already suggested crosslinks as rupture candidates. We now explicitly mention this upfront and also more clearly state the key contributions of each of these studies in this regard.

2. Later in the introduction the claim is made that collagen has evolved to 'self-report on rupture while preventing further unspecific damage'. There is no further discussion of this hypothesis nor is any evidence provided in support. I agree that the findings hint at a well-designed system, but that this is due to evolution needs to be evidenced.

We agree that at this point this is rather a proposal. To make this more clear, we have adjusted the wording at that passage and moved the more speculative statement to the

discussion.

In fact, a number of facts speak for a well-designed system: First, where collagen primarily ruptures, redox-active residues and DOPA-precursors are enriched. Secondly and more importantly, this enrichment is conserved in collagen, both throughout species [2] and across different types ($\alpha 1/2$ in types I, III, and V, as shown in new Suppl. Fig. 14). Thirdly, a recent preprint from our group, submitted during this revision, experimentally confirms the presence of DOPA in collagen of three different tissue types [19], again primarily nearby crosslinks, i.e. rupture sites.

We have added this piece of context to the respective passage. As collagen is traditionally mostly known for its mechanical properties and not the resulting (mechano-)chemistry, we think that this wider context of our findings is important to be brought up at least in the discussion.

3. The calculations of bonding energies are interesting, but require context. If we consider a purely thermodynamic event and look at equilibrium probabilities, the reported energies are key. However, bond ruptures due to mechanical forces are likely a non-equilibrium event.

We agree that the thermodynamic perspective alone is not sufficient, which is why we combine the obtained bond dissociation energies into the KIMMDY scheme. It combines the initial Morse potentials with the force distribution in the system by calculating effectively lowered barriers based on the bond elongations / bond forces throughout the system [12] in a Bell-Evans-type model along the lines $V_{\text{effective}} = V_{\text{initial}} - F_{\text{bond}} * \Delta x$. Thus, bond rupture rates depend on both, the thermodynamic bond stability as well as the force driving the rupture.

In fact, the corresponding experiments [2] operate in a quasi-equilibrium, in which the tendon is stretched at a constant load and constant extension. In this macroscopic sub-failure regime, force equilibrates (relatively instantaneously) within the structure, while bonds rupture and radicals are generated as a rare event on a seconds-to-minutes timescale. KIMMDY operates in this stretched quasi-equilibrium when we pull with a constant force setup (force equilibration occurs in the simulated systems within 20 ns, see new Suppl. Fig. 12). While this setup is commonly used in tensile tests, it is also thought to correspond to one of the most common 'use cases' of collagen tendons, which are strongly loaded when exercising but of course most of the time do not macroscopically rupture. We agree that KIMMDY in its current setup is limited to constant loading, or more generally, to loading/unloading on timescales beyond the rupture kinetics. See also our answer to 5) for some more details on the reasoning of the simulation setup.

Indeed, this important context and also the method explanation should have been pointed out more clearly and we have added context and details to the results section and to the methods as well as to the beginning of the discussion with an explicit note to the limitation that we limit ourselves to a quasi-equilibrium scenario and do not consider non-equilibrium failures of the whole fiber.

4. It has been reported that collagen backbone arrangements change significantly with mechanical forces and so do the corresponding configurations of amino acids in collagen. Some of this work is cited by the authors, though not in this context. How are such effects accounted for in this work? It seems to me that the authors explicitly exclude internal friction

and uneven distribution of forces, although this is likely what would be encountered in collagen tissues.

We agree that the uneven distribution of forces, friction and rearrangements are important and would like to point out that there are two ways we take this into account - at least to the degree that our atomistic model system can do this: First, KIMMDY takes the forces from the MD simulations as input that effectively lower the energy barriers for the kMC step, in a Bell-Evans-model type fashion [12] as described in the answer to the previous point. In the MD, the backbone can of course substantially re-arrange and certain bonds are significantly more loaded than others. This is why in Fig. 4a) backbone bonds nearby crosslinks, that have the same thermodynamic bond energies as backbone bonds elsewhere, have a higher rupture propensity compared to other backbone regions. This stress concentration dissipates, due to friction within and between the helices, along the helix.

Secondly, on a higher structural level, forces will unevenly distribute due to dislocations, flaws, etc, and some regions will be loaded more strongly than others. We attempted to account for this heterogeneous stress distribution by one of our loading setups ('shear loading'), in which some triple helices are more loaded than others (see Methods section about Molecular Dynamics simulations). Reassuringly, the major results and conclusions are the same, with ruptures funneled to the crosslinks (Suppl. Fig.7c).

We have clarified these points: We added another sentence in the KIMMDY method section to clarify the force input into rates, and now discuss our findings in the context of previous studies on protein rearrangements and friction in the Discussion section.

5. Related to these points is the question about instantaneous bond rupture vs. slow loading. Intuitively, I would assume the former is more likely, yet the authors seem to look more at the later, as they consider slow loading, distribution of forces across the fibre etc. What are the reasons for this setup?

We thank the reviewer for this important point, and also refer to our answer in 3.) in this context.

More specifically to this point, we have added a new Suppl. Fig. 12 showing the end-to-end distances of our simulated fibrils, in which it can be seen that the initial stretching happens on a 10-20ns time scale, so even if forces are only applied for some seconds to a tendon, this on a microscopic time scale equals rather a new, stretched (quasi-) equilibrium process. We note that force distribution itself through proteins (or, in fact, soft matter) occurs on an even shorter timescale, namely at the speed of sound (see e.g.

<https://journals.plos.org/plosone/article?id=10.1371/journal.pone.0064746>).

We now point out in the discussion why we consider the case of rupture at constant (equilibrated) load.

6. The authors discuss the impact of different amino acids somewhat, but then conclude that they don't actually matter despite revising their model to account for different amino acids. However, this seems to rely on the fact that the bond energy differences are very small, and this is what is used in the KMC simulations. However, this again does not account for potential alternative configurations or non-equilibrium effects. It is also in contradiction to the preference in of force accumulation in specific bonds in the back bone, partly due to the role played by Pro residues, as found in the semi-empirical calculations cited by the authors.

There are two aspects to this question, partially also pointed out already in the answers to 3)-5).

First is, as you say, that we found thermodynamic bond dissociation energies to be comparable within amino acids and, for this reason, decided to average them. In the kMC simulation, this is however only the initial barrier height.

A second input to the kMC is the force in the bond (calculated from the MD simulations) that reduces the individual barrier height and leads to bond rupture rates that are indeed varying based on the configuration (bond stretching) in the MD. As bond elongation is only a one-dimensional reaction coordinate with some limitations, we also did a detailed investigation there (this is the last section in the SI) using simple amino acid chains as a baseline. Even in this simple linear setup, we found different stretching for the same outside force. If one decided to discard this as a force field artifact and get a correction off-set for each amino acid from this (which we only did for comparison in the Suppl. Fig. 13), we would see a more smoothed out behavior with respect to amino acids but a similar concentration of ruptures concerning certain regions. The resulting discussion in the Supplement around Suppl. Fig 13 points to the fact that in any case the major conclusions of our paper are not affected by this: Whereas the rupture concentration can change locally, even if we over-corrected in the Suppl. Fig. 13, the concentration in and nearby crosslinks remains.

Inspired by the analysis of the mentioned preprint using semi-empirical calculations and your remark, we now added the rupture distribution along the GLY-X-Y pattern to the manuscript, which is 0.7% - 82.2% - 17.0% in backbone bonds (excluding the common crosslink ruptures) in our data. This is very much in line with their calculations, as they in particular for the rupturing regime report a preference for the X-position. However, in our data we did not find many prolines within these X-positions ruptures. A possible explanation is that our data is based on sampling of the fibril in which stresses concentrate in certain regions. For example, in the vicinity of crosslinks (i.e. at the end of the triple helices) stresses concentrate and ruptures accumulate, but Pro at X-positions are rare.

7. How do the authors test convergence in their KMC simulations?

For the MD simulation that serves as input in the kMC step, we have added the aforementioned Suppl. Fig. 12 showing the convergence of the fibrillar end-to-end distances of the respective simulations. This means that the bond elongations and the obtained forces in the bonds are converged as well. As another overall convergence test of the MD simulations, we varied the pull setup, of which the results are shown in Suppl. Fig. 7: Pulling from the left or right side only (instead of from both sides) does not change the results (within the noise), which would be the case if the forces needed more time to propagate throughout the system.

For the individual rates, we already tested in the original KIMMDY paper [1] that sampling of 5,000 distances per bond leads to convergence of the individual rates.

We repeat a kMC step 10,000 times for each simulation and get our rupture distribution from there, ensuring convergence of the Monte Carlo sampling. We only invoke the first (or second) rupture and do not have any convergence of KMC in the classical sense that all states have to be visited.

We now more thoroughly discuss convergence in the methods section.

8. Line 348 - the total accumulated simulation time is meaningless. Total simulation time has no relation to convergence or sampling and should not be used.

We agree that the total simulation time might be mistaken with convergence and have removed it. To still summarize the scope of the study, we now instead mention that more than 70 individual 100 ns MD simulations of different systems and their replicas were carried out.

Reviewer #2 (Remarks to the Author):

This is an interesting paper providing novel detailed analysis on the mechanics of collagen, and bond rupture propensity therein. While the calculations and comparison of different bond strengths is novel, this reviewer has several concerns about suggested multiscale mechanisms that govern bond rupture in mature fibrils.

We thank the reviewer for the overall positive evaluation and will address the concerns in the following.

MD simulations provide a basis for KIMMDY simulations, however, the equilibration times before pulling simulations are very short (10 ns).

We agree that sufficient equilibration is important and like to note the following points

i) Our interest was to appropriately sample the stretched state, and we thus focused on that one for this demanding multi-million system. Prior to stretching, it is 10ns each for NVT and NPT, which has been suggested as sufficient to enable stable simulation conditions.

ii) More importantly, however, the 100ns pulling simulations represent another equilibration to reach the stretched state: Only thereafter, we sample the bond elongations for KIMMDY with another short MD run.

Again, we kept the equilibration in absence of force short, as our interest here is collagen under tension. This MD setup proved useful also previously [2].

We have also added a new Suppl. Fig. 12 (that is also discussed in the answers to Reviewer#1) showing the convergence of the end-to-end distances of the fibrils as a measure that we have simulated/equilibrated long enough under force to reach a quasi-equilibrium stretched state before sampling the bond elongations.

The authors discuss that the multiscale topology of collagen determines how force is distributed throughout the structure and thereby modulates bond rupture. Based on this supposition, how is the location of crosslinks identified in the models? Is the selected location of the crosslinks consistent with the heterogeneity expected? What is the control for crosslink location? Do crosslinks differ across species and how is this captured in simulation? Furthermore, experiments are limited to a single species – can this choice be explained?

We thank the reviewer for this important question. For the atomistic models in the simulations, the whole modeling process is described in more detail in the ColBuilder

publication [25] that is the source of the models. We consider enzymatic crosslinks, but no glycation products that arise in older tissue at unspecific sites. The enzymatic crosslinks are derived from lysines only, and specific crosslink positions have been identified by mass spectrometry, see e.g. [57]. They can vary across species due to slightly different sequences. Here, we chose three species to represent that variation but did not see any qualitative nor major quantitative differences between them (compare Suppl. Fig7b). From the two main crosslink types - divalent and trivalent - the amount of which varies across tissues and age, we chose one each to represent this heterogeneity as well. As the manuscript points out, the choice of crosslink type can impact the first and subsequent rupture sites. Given the computational resources, we think this is a good compromise to sample the heterogeneity among species and enzymatic crosslink on the microscale. Coarser models will be needed to exhaustively sample other crosslinks, species, and mixtures of crosslinks types at varying mol/mol. We now discuss this as an outlook in the Discussion.

Three observations support that other crosslink topologies likely give analogous results. First, for biological tissue, it is known that crosslinks differ not only between species but also across tissue type. For example, tail tendon has significantly less pyridinoline cross-links compared to other tendon sources. We originally chose to investigate rat tail tendon as a model for other tendons as it is a readily available tendon source that has been exhaustively studied in the field of collagen biology. However, following the concern of the reviewer, we now also investigated achilles and flexor longus tendon from rat. We could reproduce the shift in molecular weights upon mechanical stress application (new Fig. 5e and Suppl. Fig. 10). Again, stressing causes the loss of dimers and the increase in fragments of lower mass than collagen alpha1 or alpha2 monomers.

Secondly, as a response also to Rev #3, we compared in a sequence alignment of collagen I, III, and V the neighborhood of the crosslinks as primary rupture sites according to the simulations, and find radical scavenging residues to be strongly conserved around the (also conserved) crosslinks (new Suppl. Fig. 14). This is indirect evidence that our conclusions can be generalized also to other collagen types. In the future, we aim to expand our study across other species.

Thirdly, we also like to note that a similar increase in EPR signal upon pulling across different tissue types (achilles, tendon, meniscus) has now been published as a preprint from our group [19], indicating that beside these differences we have a similar underlying rupture process.

Authors discuss primary and secondary rupture, in particular in the context of trivalent crosslinks. They explain that “the total bond rupture rate in the damaged trivalent crosslink decreases by ~3 orders of magnitude compared to the pre-rupture simulations, indicating that stress in this crosslink overall relaxed after its rupture.” Can the authors measure this stress distribution directly to support their claim of the mechanism? There appear to be different contributing mechanisms to rupture – bond strength and force distribution in the vicinity of the bond based on the topology of the bond location. The rupture rate does not directly explain which mechanism is responsible.

This is an interesting point. The bond strength did not change in comparison to the undamaged case, so it has to be due to stress distribution. To look into this mechanism, we measured the extension of the other extended arm in Suppl. Fig. 8: As you can see there, it

can use the previously hidden length, which in turn means a lower stress. Thus, stress redistribution allows the connection mediated by the originally trivalent crosslink to be maintained. We tried to clarify this further in the result section.

The experimental results support both breaking of backbone and crosslink bonds, but does not compare their breaking propensity. As a result, it is not clear that the experiment confirms the trends observed in the models.

Indeed, the experimental results have been interpreted to support both backbone breakage (peptide fragments) and altered cross-linking (reduced densitometry of higher molecular weight chains), but only qualitatively. They confirm both mechanisms to be present but cannot make a quantitative statement. We have added a sentence to the manuscript at the end of the results section explicitly pointing to the limitation that their relative prevalence remains open for future work.

We also like to note that as we discuss in the manuscript for the simulation results, the reported competition between backbone vs. crosslink can depend on parameterization, set-up etc. But for a wide range of parameters (i.e. a lowest BDE of 220 kJ/mol vs 282 kJ/mol in the PYD crosslinks), we always at least recover this competition, so the experiments are in line with that overall key result.

Can the authors specify in their methods (or SI) further details about the parameterization within KIMMDY from calculations? Specifically, for multi-armed crosslinks, the mechanical response may differ along different directions; is this captured in the KIMMDY scheme?

Following the comments of reviewer #1, we have already added some further details that could have been indeed more clear. We have now added that bonds are modeled with Morse potentials, of which the width and equilibrium bond length are taken from the regular force field parameters. The initial barrier height given by the BDEs is effectively lowered by the work that acts on the individual bond, leading to a new, effective barrier that depends also on the force in the bond (i.e. on its average elongation in the MD).

With this, the loading of the multi-armed crosslinks does not need to be parametrized specifically: As for any other bond in the system, we use their pre-computed BDEs (Fig. 2a), and the loading happens naturally in the MD simulation, which is why we can easily capture different responses including the multi-armed crosslink.

Reviewer #3 (Remarks to the Author):

This exciting manuscript provides a highly interesting and original perspective on the question how the hierarchical structure of collagen contributes to the remarkable mechanical properties of this protein. Traditionally this problem has been thought of purely in mechanical terms, but this work demonstrates that in addition there is a unique mechanochemical mechanism at work. By advanced scale-bridging simulations, the authors convincingly show that the structure of collagen is set up such that trivalent crosslinks can serve as sacrificial bonds and also such that mechanoradicals created by covalent rupture are quickly stabilised. The simulation predictions are qualitatively supported by experiments where SDS-PAGE and mass spectrometry is used to analyse the fragments liberated by rat tail

tendon upon mechanical stress. The analysis of these experiments should be clarified. Also, the authors should clarify the generality of their findings for different collagenous tissues and animal species.

Thank you for that very positive assessment, indeed the mechanochemical coupling in collagen is a perspective we are very excited about.

Comments

1. The experiments in Fig 4 show corroborating evidence that is consistent with the simulations, but do not provide a true, stringent test of the predictions. A key prediction (Fig 4) is for instance the key role of trivalent crosslinks, which the authors could test experimentally by either comparing rupture patterns (i.e. fragment fingerprints) for collagenous tissues that differ in trivalent crosslink content (e.g. because of different age), or by comparing rat tail tendon against in vitro-assembled collagen fibrils, which only have divalent crosslinks.

We agree that there are many more ways to strengthen this picture.

First, we have now generalized our findings to different tissues: In a completely new experimental data set added during the revision, we now compare rat tail tendon to flexor and achilles tendons. Compare Fig. 5 with a new panel e and the new SDS-PAGE gel shown in Suppl. Fig. 10.

Overall, we can confirm both the increase in lower mass fragments (indicating backbone rupture) and a reduction in oligomers (indicating crosslink rupture) in all of the data sets. Albeit the higher amount of trivalent crosslink expected for achilles, we did not see a stronger drop in the dimer fraction compared to tail tendon. A possible explanation is that rupture at the sacrificial bond in one arm of the trivalent crosslink would still leave the linkage between two collagen chains intact. However, we also emphasize that variations across tissues are large, hampering direct comparisons at these sample sizes.

Secondly, we additionally attempted to compare rupture across collagenous tissues of different age, for both achilles and tail tendon. Again, such an analysis is challenged by the large scatter when working with native tissue. Still, the preliminary data shown below suggests that during aging, less ruptures occur (as quantified here by EPR intensity, i.e. amount of radicals) at a given external force applied (Figure a and c for tail and achilles, resp). At the same time, stiffening caused by aging results in lower extensions at a given force (b and d for tail and achilles, resp.). A conclusive analysis requires a larger age range, more statistics and another loading scheme (constant extension), and will definitely be followed up in the future.

Fig. Stretching experiments in rat tendons. For the experiments, samples were isolated from rat tail and Achilles' tendons. Specimens equilibrated to atmosphere (~25 °C/50% hum.) were stretched for 1000 s at 15 N(const.). Before and after the stretching experiment, the content of organic radicals was quantified by Electron paramagnetic resonance (EPR) spectroscopy by the peak height at g -factor=2.005. (a, c) EPR signal increment determined for tail (a) and achilles tendon (c) as ratio of peak height before and after stretching. (b, d) Extension of the sample at given force calculated in percent, for tail (b) and achilles tendon (d). Red dots/bars represent mean and standard deviation of the respective age group, gray points show individual values for one tendon stretching experiment.

Thirdly, to our understanding in-vitro assembled collagen does not harbor crosslinks, neither divalent ones, at least not native crosslinks in any quantitative amount. In fact, in our very early studies before moving to native collagenous tissue, we attempted to measure radicals in reassembled collagen but never could detect mechanoradicals after mechanical perturbation. We attributed this absence of radicals to the lack (or small amount) of crosslinks overall, and moved on to native tissue, which can be subjected to much higher loads and readily shows mechanoradicals already in the subfailure regime.

2. Regarding the experiments:

a. the authors should clarify what is the nature and magnitude of the mechanical stress, which is unclear from the described procedure (dried fibers were crushed, cooled and filled with liquid nitrogen?), and whether or not this is representative of the effects of mechanical stretching they explore in the simulations.

We have crushed the original set of fibers with a mortar, which means that the stress level is above the simulation setup. This is necessary to obtain a signal in the experiments, but also justified as we have seen previously in EPR experiments that the type of signal, i.e. the created radicals, remained the same in crushing as in more mild stretching with an extensometer [2]. We agree that this is important information and have added a sentence to the method section. During the revision, we have added more experimental data to generalize to other tissue type as aforementioned, but have also changed the protocol of stressing the fiber, now using ball-milling. This is a quantitatively more controllable method (in terms of speed and frequency), for which we also have added a paragraph in the methods.

b. The authors use rat tail tendon. What is the age and crosslinking pattern for this tissue? Typically tendon has more immature crosslinks than other tissues such as dermal tissue.

The rats were 5 months old. Indeed, tail tendon collagen is known to harbor more immature divalent crosslinks and less so mature trivalent PYD crosslinks. Prompted by this and the earlier comment of this reviewer as well as a related question of Reviewer #2 (point 2), we now measured the molecular weight distribution also for the more relevant case of achilles and flexor longus tendon tissue, and confirmed the previous observation. Experiments with in-vitro assembled collagen and with tissue from slightly different age groups have so far not been conclusive. See our answer above for details and additional data (point 1.). For the newly added second experimental set, we added a paragraph to the methods section.

Dermal tissue is definitely on our list of interesting samples to test next, but we consider it out of scope of the present study given its complex architecture and composition (e.g. type I as well as III). We also note that access to rat tissue of very different age groups has so far been a challenge.

c. Why are there collagen fragments found for unstressed collagen?

Even before the mechanical treatment, there is some unavoidable stress to the fibers in preparation: They need to be cut out, washed etc., such that some small amount of fragmentation might result from this. We know a similar behavior from previous studies [2], in which a smaller, but non-zero, signal in an EPR indicated radicals also in unstressed fibers. Also note that the SDS-PAGE samples were overloaded. Therefore, although a small amount of potential degradation was observed in the control samples, it was not a statistically significant amount in comparison to the amount of non-degraded alpha chains. In the data set added during the revision, we have found even more unexpected lower mass bands, but found them to be other (muscle) protein via a mass spectrometry analysis, as described in the added Suppl. Fig. 10.

d. Why is there strong variability for the 4 samples in Fig 5a (right)?

The variability arises due to the aforementioned overloading and the fact that the stressed sample uptake is less controllable. In the new data set, we improved upon this and achieved an approximate equal signal strength in the monomer bands. Again, we would like to note that we here work with native tissue, i.e. with samples of intrinsic variability.

e. The authors should improve the annotation of the data in fig 5. It is unclear from the figure which bands from (a) are seen in (b) (arrows could be used) and which bands are analysed (separately or pooled?) in c and d.

Thank you for the suggestions, we have edited the figures along them and now also added a new panel e with the new data set.

We have pooled all the bands for the sequence coverage analysis in c) for clarity. However, the individual sequence coverages can also be found in Suppl. Fig. 9, to which we now point more clearly. In contrast to the bands in c), in panel d) we measure the signal intensity of all fragments below the monomers (so no cut-out needed), which is why we there analyzed the whole lane and no bands.

3. Data for MD simulations with 3 different collagen sequences are pooled together in Fig. 3. Why were the data pooled? And did the analysis for different collagen sequences show any differences among the 3 species?

Indeed, the species differ in sequence, but only so much as homology modeling into the experimental fiber diffraction data that was the basis of the model was still possible – for details see the publication of ColBuilder [26]. We have analyzed them also separately and did not see any qualitative differences in the rupture pattern - compare Suppl. Fig. 7b. For this reason and clearness, we have pooled the data in the main manuscript.

4. For context, it would be helpful if the authors could discuss both the stress and corresponding strain in their simulations and how it compares to typical stresses and strains encountered in collagenous tissues.

We agree that this is a helpful context, which we have now added to the manuscript. As stated in the methods, we apply about 1nN per chain / 3 nN per triple helix, being in a high-load yet sub-failure regime. We have added Suppl. Fig. 12 showing the end-to-end distances and reach strains in the range 20-24%. This is rather on the higher side of collagenous tissues, but can be explained by the absence of AGE crosslinks that otherwise stiffen tissues. We have added this context in the result section. Another discussion point that we have now added and that was also made in [2] is that we need higher forces in the “perfect” atomistic model system than in experiments to achieve second-to-minute rupture time scales, as in reality material flaws could lead to stronger local stress concentrations and faster ruptures.

5. The rupture simulations are performed on atomistic collagen type I fibril models. Can the authors speculate how higher-order fiber structures, which depending on tissue type can be twisted or not, will affect their mechanism? And collagen fibrils in tissues are made of mixtures of collagen I and minor fibrillar collagen types (e.g. III, V), how will this affect rupture?

We speculate that at a higher structural level, twisting, flaws, glycosylation and other features can rather increase stress concentration in different areas, leading to even earlier rupture at a given macroscopic strain, which would explain the rather low stresses radicals are generated by [2]. Coarser models, such as MARTINI or ultracoarse-grained models will be needed to address these questions computationally - we have pointed to this possible extension now in the second paragraph of the discussion. However, as the building blocks of them are still similar to our models, within them more locally our findings of preferred crosslink rupture (with competition from backbone) should hold.

We also thank the reviewer for pointing us to other collagens such as types III and V. We performed a sequence analysis of these types for rat, the species we focus on here (new Suppl Fig. 14). Interestingly, some Phe and Tyr residues are either conserved, in particular across alpha1 chains, or just shifted in their positions. Thus, also III and V show clusters of aromatic residues that, in form of DOPA, could be ideal to scavenge radicals after bond rupture (see Kurth et al, biorxiv, [19]). This conservation indirectly supports the notion of ruptures at or nearby crosslinks, also in III and V. We now have added this point to the Discussion.

Minor comments

1. The authors should change the terminology by which they describe the SDS-PAGE gels; “banding pattern” in the context of collagen typically refers to collagen’s axially ordered D-banded fibril structure.

Thank you for pointing out this ambiguity, we have changed the terminology depending on context to make the differentiation more clear.

2. The authors should clarify their statement at the bottom of p2 “SDS-PAGE coupled to MS supports altered collagen crosslinking” (altered as compared to what?)

We have clarified that the crosslinking is altered in stressed samples compared to the control.

3. The authors should clarify their statement on p7 “The weaker the crosslinks are compared to the backbone, the more they can prevent any other unspecific position in the protein backbone.”

Thanks for pointing to this, we have added some words that were missing, especially that they prevent first ruptures at other backbone positions.

4. The authors should clarify what they mean by “as a measure of collagen crosslinking quality” on p10.

We have changed the wording to express more clearly: Our experiments support clearly a qualitative change (e.g. potential changes in the crosslink usage in the tissue). In contrast, we cannot necessarily make a more quantitative statement (e.g. about the concentration of moles/mol of a specific crosslink).

Reviewers' Comments:

Reviewer #1:

Remarks to the Author:

The authors have addressed all points I have raised about their initial manuscript. Their responses have been detailed and the resulting changes to the manuscript have improved the presentation of their work and added important information. In my opinion, the manuscript is now suitable for publication, and I have no further queries.

Reviewer #2:

Remarks to the Author:

My comments have been sufficiently addressed. This is an exciting manuscript and I congratulate the authors on a great piece of work.

Reviewer #3:

Remarks to the Author:

The authors have satisfactorily addressed all reviewer comments, adding important new data and clarifications.